# CONV-COA: OPEN-DOMAIN QUESTION ANSWERING VIA CONVERSATIONAL CHAIN-OF-ACTION WITH HOPFIELD RETRIEVER

## ABSTRACT

We present a Conversational Chain-of-Action (Conv-CoA) framework for Open-domain Conversational Question Answering (OCQA). Compared with literature, Conv-CoA addresses three major challenges: (i) unfaithful hallucination that is inconsistent with real-time or domain facts, (ii) weak reasoning performance in conversational scenarios, and (iii) unsatisfying performance in conversational information retrieval. Our key contribution is a dynamic reasoning-retrieval mechanism that extracts the intent of the question and decomposes it into a reasoning chain to be solved via systematic prompting, pre-designed actions, updating the Contextual Knowledge Set (CKS), and a novel Hopfield-based retriever. Methodologically, we propose a resource-efficiency Hopfield retriever to enhance the efficiency and accuracy of conversational information retrieval within our actions. Additionally, we propose a conversational-multi-reference faith score (Conv-MRFS) to verify and resolve conflicts between retrieved knowledge and answers in conversations. Empirically, we conduct comparisons between our framework and 23 state-of-the-art methods across five different research directions and two public benchmarks. These comparisons demonstrate that our Conv-CoA outperforms other methods in both the accuracy and efficiency dimensions.

## 1 INTRODUCTION

We propose a conversational reasoning-retrieval framework to enhance the efficiency and quality of OCQA, tailored to surpass the architecture of Retrieval Augmented Generation (RAG) methods. This work addresses three major challenges in applying RAG to answer conversational questions: (i) **weak reasoning**, where large language models (LLMs) struggle to acquire required information from heterogeneous sources, (ii) **unfaithful hallucinations**, where the response may not align with real-time or domain-specific facts, and (iii) **unsatisfying retrieval**, where the traditional dense information retriever (IR) cannot get the intent of questions and fails in the conversational scenario.

To enhance the reasoning, faithfulness, and conversational IR, previous approaches such as chain-of-thought-based work (Saparov & He, 2022; Yao et al., 2023a; Xiong et al., 2024) prompt LLMs to answer complex questions step by step. The other work proposes RAG-based prompting frameworks, such as agents (Pan et al., 2024). However, they aim to solve single-round questions and fail in complex conversations. More recent work focuses on the retrieval phase and aims to improve the query quality and retrieval capability. While CONQRR and ReExCQ (Wu et al., 2021; Mo et al., 2023b) are the representative query-reformulation methods to rewrite and expand the current query with historical conversations, they need extra pre-trained model with much training expense. We argue that training a model for query reformulation is unnecessary; instead, prompting methods can harness the existing LLMs to serve this purpose. ConvSDG and CONVAUG (Mo et al., 2024b; Chen et al., 2024) try to enhance the conversational retriever by generating different levels of conversations. However, their performance cannot be beyond the theoretical upper bound of traditional dense retrievers, and their consumption is still expensive. In summary, the key challenge of current OCQA lies in designing a framework tailored for conversational QA scenarios that integrates prompting with RAG and devises an innovative conversational retriever that surpasses traditional retriever architecture in efficacy.

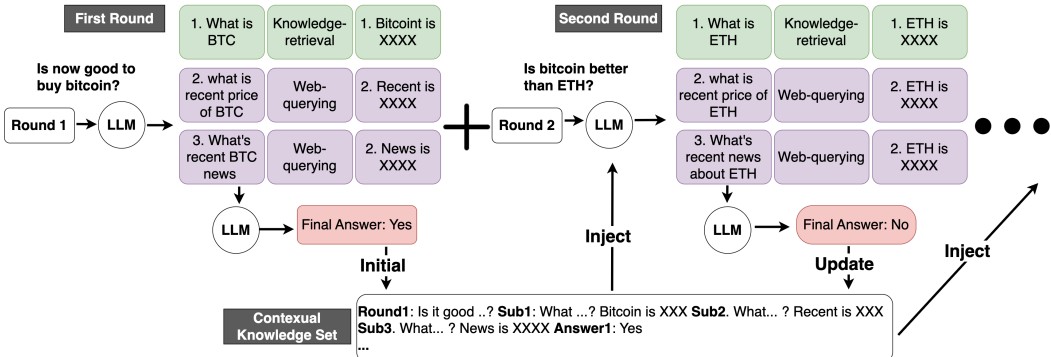

Figure 1: **Overview of Conv-CoA.** It starts by injecting the initial question into a prompt, creating an action chain (AC) via the LLM. Each node in the AC represents a sub-question and an initial answer. Specific actions verify if the initial answer needs modification based on retrieved data. If the confidence in the initial answer is lower than in the retrieved data, an action prompts a change. The AC contents are stored in a CKS and updated at the end of each turn. In future turns, the CKS and the current question are combined to regenerate the AC, using different prompt templates to generate sub-questions for unknown content, thus reducing information overlap.

To this end, we propose **Conv-CoA** framework, to deliver an accurate, reliable, and efficient OCQA. Conv-CoA introduces two predefined actions: web-querying and info-searching. The former retrieves the latest information from the Internet, while the latter accesses knowledge from storage (i.e., specific domain documents). Since both actions reply on effective data retrieval, we design a resource-efficient Hopfield Retriever, inspired by the Modern Hopfield models showcasing fast convergence and exponential memory capacity (Schimunek et al., 2023; Widrich et al., 2020). This retriever surpasses traditional dense retriever architectures in training speed and inference precision. Meanwhile, to minimize unnecessary time and token consumption, we design a systematic prompting mechanism to decompose a complex question into a reasoning chain with necessary sub-questions that need external help. As shown in Figure 1, the first round question is embedded into a designed prompt to construct an action chain (AC) using the LLM. Each node in the AC represents a sub-question paired with an initial answer generated by the LLM. We assign actions to verify if the initial answers align with unprepossessing retrieved data by proposed conversational-multi-reference faith score (Conv-MRFS). If a discrepancy arises, the action refines the answer using summarized content. This mechanism avoids unnecessary processing which is the main part of cost in RAG, significantly improving efficiency. The refined AC content is stored in a contextual knowledge set (CKS), which is updated at the end of each future round. At the same time, the LLM generates the final answer based on the updated CKS. For subsequent rounds, Conv-CoA iteratively combines the CKS and the current round's question to repeat the AC generation via a different prompt template. We design a rule to generate the sub-questions that we do not have relevant content yet based on the newest CKS to minimize the latency of unnecessary retrieval.

In summary, Conv-CoA is the first work that enables a faithful, accurate, fast OCQA by incorporating the prompting method and the novel Modern Hopfield retriever. Our main contributions are as follows:

- Conv-CoA framework can decompose query into necessary sub-questions to solve by Hopfield-enhanced actions with verification of Conv-MRFS score.
- We propose a resource-efficient Hopfield retriever to break the limitations of traditional retrievers and significantly speed up the training with even better accuracy.
- We design a memory-enhanced mechanism to construct each round's AC based on the updating CKS, enabling less overusing retrieval to further reduce the cost.
- Experiments demonstrate that we outperform other methods and solve three challenges.

## 2 RELATED WORKS

To improve OCQA performance, recent work focuses on RAG methods. We categorize these methods into two distinct phases in alignment with the RAG life cycle: (1) Retrieval Phase: methods aim to enhance query quality and retrieval efficacy, facilitating the acquisition of more pertinent information to support LLM in generating answers. (2) Generation Phase: methods aim to propose prompting methods to aid LLM in conducting reasoning processes, thereby enabling more rational and precise answer generation. Hence, we first introduce the work in the retrieval phase from two perspectives:

query quality and retriever capability. Then, we explore various prompting methods designed to synergize with the retrieval phase. In addition, we also introduce some Hopfield-based methods that motivate us to propose our memory-enhanced conversational retriever.

**Query Reformulation Methods.** Conversational Query Reformulation (CQR) uses query rewriting and expansion based on a conversational context to improve retrieval performance. Compared with other conversational retrieval methods, CQR directly reformulates the original conversation-based query into a standalone query as input to off-the-shelf retrievers without fine-tuning. While previous CQR work addresses conversational retrieval by using human-rewritten queries or querying expansion methods (Lin et al., 2021; 2020; Mo et al., 2023a), they always get a sub-optimal and require a separate model trained by lots of human rewrites. To address these drawbacks, CONQRR (Wu et al., 2021) optimizes the query rewriting model to the retrieval. More recently, IterCQR and IQR (Jang et al., 2023; Ye et al., 2023) conduct query reformulation without relying on human rewrites by prompting the large language models (LLMs). ReExCQ (Mo et al., 2023b) focuses on expanding the current query with selected relevant historical queries. However, they need a large storage capacity and time when generating candidates during the training.

**Enhanced Retrieval Methods.** Conversational Retrievers train previous information retrievers on conversational datasets using more complicated training strategies and loss functions. It aims to enhance the retriever's ability to search for relevant information within conversational situations. LeCoRE (Mao et al., 2023) considering knowledge distillation, InstructorR (Jin et al., 2023) utilizing LLMs to predict the relevance score between the session and passages, and SDRConv (Kim & Kim, 2022) that includes mining additional hard negatives. In addition, ConvSDG and CONVAUG (Mo et al., 2024b; Chen et al., 2024) are the most recent work about utilizing LLM to generate conversations. ConvSDG explores the dialogue/session-level and query-level data generation separately. CONVAUG generates multi-level augmented conversations to capture the diverse nature of conversational contexts. HAConvDR (Mo et al., 2024a) incorporating context-denoised query reformulation and automatic mining of supervision signals based on the actual impact of historical rounds. However, all of them focus on augmenting the training data for conversational dense retrievers. Despite these advancements, the performance upper limit of these methods still cannot break through the theoretical upper limit of transformer-based dense retrievers.

**Hopfield Models and Dense Associative Memory.** Classical Hopfield models (Hopfield, 1984; 1982) are energy-based physics models emulating the human brain's associative memory, emphasizing memory pattern storage and retrieval. Modern Hopfield models (Krotov, 2023; Hu et al., 2024c;a;b; 2023; Wu et al., 2024; 2023; Krotov & Hopfield, 2021; Ramsauer et al., 2020; Krotov & Hopfield, 2016), also known as Dense Associative Memories, are advanced versions of the classical Hopfield networks. They offers improved memorization capacity (Hu et al., 2024c; 2023; Wu et al., 2024; Krotov & Hopfield, 2021; Ramsauer et al., 2020; Demircigil et al., 2017; Krotov & Hopfield, 2016) and compatibility with transformer architecture as advanced attention mechanisms (Hu et al., 2024c; Wu et al., 2024; 2023; Schimunek et al., 2023; Auer et al., 2024; Hu et al., 2024a; 2023; Widrich et al., 2020; Ramsauer et al., 2020). We propose a Hopfield-based retriever for conversational search, leveraging the rapid convergence and vast memory capacity of modern Hopfield models to better retrieve knowledge from memory spaces.

# 3 CONVERSATIONAL CHAIN-OF-ACTION

Our framework aims to generate answers aligned with the current conversational question, denoted by $q_n$ for the $n$-th round. This QA process is historically contextual, leveraging previous dialogue rounds denoted by $\mathcal{H} = \{q_i\}_{i=1}^{n-1}$. It is essential for the framework to optimize the formulation of each question $q_i$ to accurately capture the user's intended query content $\mathcal{O}_n$. Utilizing the reasoning capabilities of LLMs, the framework decomposes the optimized question into a chain of $k$ sub-questions $\{\mathcal{R}_{n1}, \mathcal{R}_{n2}, \ldots, \mathcal{R}_{nk}\}$, each aimed at a specific aspect of the main query. For each sub-question $\mathcal{R}_{ni}$, the system retrieves the most relevant information passage $\mathcal{I}_{ni}$ from a corpus of external data sources, culminating in a set $\{\mathcal{I}_{ni}\}_{i=1}^{k}$ that aids in generating the final answer $a_n$. Thus, the abilities to optimize questioning $\mathcal{O}$, to chain reasoning $\mathcal{R}$, and to retrieve pertinent information $\mathcal{I}$ are pivotal. The goal of this paper is to propose a framework that integrates disparate external information sources, empowering the LLM to dissect queries effectively and retrieve related content swiftly, leading to the provision of the most precise answer $\mathcal{A}_n$.

## 3.1 OVERVIEW

As shown in Figure 1, we utilize in-context learning to inject the 1st round query into a prompt, generating an Action Chain (AC) via an LLM. Each node of AC includes sub-questions paired with initial answers. The assigned actions then verify if the initial answers align with non-post-processing retrieved data by Conv-MRFS. If any answers fail, corresponding actions continue to summarize content for adjusting them, avoiding unnecessary post-processing, which costs a major part of RAG. Meanwhile, we maintain and iteratively update a Contextual Knowledge Set (CKS) at the end of a conversation round, which stores essential information of refined AC and reduces redundancy in retrieved data. This dynamic updating supports the decomposition of subsequent queries into necessary sub-questions that have no relevant information yet. It helps to minimize the latency of unnecessary retrieval. In the retrieval phase, our proposed Hopfield-based retriever ensures more powerful conversational retrieval, enriching the contextual understanding and response precision.

## 3.2 CONTEXTUAL KNOWLEDGE SET

The Contextual Knowledge Set (CKS) is a structured data format designed to store critical information from each round of a conversation. CKS is in JSON format, which facilitates easy integration and manipulation in data-driven applications. Each entry in the CKS records various components of the dialogue, including the original question posed during the round, the optimized question that refines or extends the original inquiry, detailed sub-questions that break down the main question into more reasonable parts, summarized information that has been retrieved in response to each sub-question, and the final answer concluded at the end of the discussion. This structure not only preserves the flow of conversation but also enriches the contextual understanding of each interaction. Below is an example of how CKS organizes and presents this information:

```
{ "Contextual knowledge set": [
    {
      "round": 1,
      "original_question": "What ...?",
      "optimized_question": "...",
      "sub_questions": {
        "sub1": "What are ...",
        "sub2": "What are ...",
      },
      "information_summaries": {
        "infor1": "A use light ...",
        "infor2": "B are the ...",
      },
      "answer": "Photosynthesis ..."
    }, ...]
}
```

Listing 1: Example of Contextual Knowledge Set JSON Structure

## 3.3 ACTION CHAIN GENERATION

We have two different action chain generation stages, the initial and normal stages, aiming for the first round and future rounds in a conversational question answering.

**Initial stage:** In this stage, we inject the first round questions and detailed descriptions of actions into a prompt template. Then, LLM decomposes the question into sub-questions paired with guess answers and assigned actions. The prompting template is shown in Appendix E. After that, each assigned action retrieves the related information and checks if the guess answer is not aligned with the unprepossessing information by the Conv-MRFS score to determine whether to adjust these sub-questions. Finally, LLM generates the final answer from the processed action chain, and the framework updates the CKS with the refined action chain.

**Normal stage:** In the normal stage, we design a different prompt to promote LLM to extract the N-round intent question from the updated CKS and original N-round question, then decompose it into necessary sub-questions that have no related information yet. After that, actions do the retrieve and answers correction again. Finally, LLM generates the final N-round answer and updates the process into the CKS iteratively. The prompting template is shown in Appendix E:

### 3.4 ACTION IMPLEMENTATION

We propose two actions to address different information needs: (1) Web-querying, which searches real-time information from the Internet, and (2) Knowledge-retrieval, which retrieves the relevant information within domain-specific corpus datasets. Both actions follow the same three-stage workflow: (i) Information Retrieval, (ii) Alignment Detection, and (iii) Answer Correction. At the core of the Information Retrieval stage for both actions is our novel Hopfield-based Retriever, which enables efficient and accurate conversational search. In the following sections, we first introduce the action designs. We then provide a detailed explanation of the three workflow stages, with a particular emphasis on the innovative Hopfield-based Retriever.

#### 3.4.1 ACTION DESIGN

**Web-querying.** Web-querying action leverages search engines, employing a query strategy to retrieve relevant Internet content. Initially, the action searches for keywords from the specified sub-question $\mathcal{R}_{nk}$, providing a result list. We select the top-k results and extract the content directly from their web pages as the unprepossessing information to detect the alignment between it and LLM-generated initial answers. If the nonalignment exists, we collect the titles $T$ and snippets $Sn$ from the top k pages. Each title and snippet pair $\{T_k, Sn_k\}$ is then transformed into a 1536-dimension vector $Emb\{T_k|Sn_k\}$ using the OpenAI's text-embedding-ada-002 model (OpenAI, 2023). We perform the same vector transformation for the sub-question and its guess answer $\{\mathcal{R}_{nk}, \mathcal{G}_{nk}\}$. Subsequently, our Hopfield-based retriever selects the most similar pages from vectors $Emb\{T_k|Sn_k\}$ by $Emb\{\mathcal{R}_{nk}|\mathcal{G}_{nk}\}$. The contents from these pages are then extracted and summarized to correct the related sub-question.

**Knowledge-retrieval.** In our framework, documents sourced from diverse platforms, including Wikipedia, are pre-processed using an encoder designed to transform textual content into embedding vectors. Each document is segmented into multiple chunks determined by their length, following which these segments are encoded into vectors. These vectors are subsequently cataloged in a vector database, indexed for efficient retrieval. For the retrieval process, we leverage our novel Hopfield retriever, which utilizes these pre-encoded vectors to respond to queries effectively, ultimately yielding the top-k most relevant results based on similarity metrics. The details of the encoding and retrieval mechanisms will be further elaborated in the subsequent section of this paper 4. After retrieval, if there exists a nonalignment between the retrieved information and the guess answer $\mathcal{G}_{nk}$, Conv-CoA prompts the LLM to summarize the retrieved results and correct the answer.

#### 3.4.2 ACTION WORKFLOW

For each sub-question, an action follows the same three-stage workflow as shown in Algorithm 1.

**A. Information Retrieval.** In this stage, the task begins by transforming the sub-question and guessed answer into a vector representation compatible with the information pool (e.g., Internet data or local domain-specific corpus databases). To achieve this, we utilize the existed embedding model to encode the query, $QS_n = \{\text{Sub}_n \mid A_n\}$ into a vector, $Emb\{QS_n\}$. This vector aligns with the embedding space of the information pool. Subsequently, using our Hopfield-based retriever, which incorporates the conversational context, we search for the top-$k$ most similar results. The final retrieved results, $R_{\{QS\}}$, are expressed as: $R_{\{QS\}} = (r_1 \mid r_2 \mid \cdots \mid r_k)$.

**B. Alignment Detection.** To verify conflicts between guess answers and retrieved information, we introduce the Conversational-Multi-Reference Faith Score (Conv-MRFS) to evaluate the consistency of guess answers with the conversational context (conversation history CKS and new retrieved information), helping to avoid unnecessary post-processing of retrieval and reduce the hallucinations.

B1. Overview of Conv-MRFS: The Conv-MRFS involves extracting relevant segments from the conversation history as references. The core is the faith score, which measures alignment between the guess answer and reference segments based on precision, recall, and average word length.

B2. Components of the Faith Score: The faith score $S$ is a composite metric incorporating three components: Precision (P), Recall (Rcl), and Average Word Length (AWL). These components are weighted to reflect their importance in the evaluation process: $S = \alpha \cdot P + \beta \cdot Rcl + \gamma \cdot AWL$. Here, $\alpha, \beta, \gamma$ are weights for precision, recall, and average word length, summing to 1 for normalization.

B3. Precision and Recall: Precision (P) quantifies the fraction of relevant instances within the generated answer consistent with the conversational context and Recall (Rcl) measures the fraction of

relevant instances from the conversational context captured by the generated answer:

$$P = \frac{\text{\# of consistent items}}{\text{total \# of items in } A_n}, \text{Rcl} = \frac{\text{\# of consistent items}}{\text{total \# of relevant items in CH}}.$$

B4. Average Word Length: Average Word Length (AWL) represents the mean length of words in the generated answer, indicating verbosity and informativeness: $\text{AWL} = \frac{\text{sum of lengths of all words in } A_n}{\text{total \# of words in } A_n}$.

B5. Score Calculation: For each segment $\text{CH}_i$ in the conversation, we compute the faith score $S(\text{CH}_i, A_n)$. The Conv-MRFS is the maximum score across all segments.

B6. Threshold Decision: We set a threshold $T$ for answer. If the Conv-MRFS exceeds $T$, the answer is considered faithful. Otherwise, the answer is revised to better align with the retrieved information.

**C. Answer Correction.** If a misalignment is detected, the retrieved information $R_{\{QS\}}$ and the corresponding sub-question $\text{Sub}_n$ are fed to the LLM. The LLM generates a corrected answer $A'_n$ and a reasoning summary that links retrieved data to the correction. The corrected answer and rationale are then stored in the CKS for future rounds.

### 3.5 CONV-COA-PLUS

To further enhance the structured reasoning ability of LLMs, we propose **Conv-CoA-Plus**, an extension designed for open-source LLMs. Specifically, we incorporate a reinforcement learning module that fine-tunes the model using GRPO (Shao et al., 2024). Detailed descriptions of Conv-CoA-Plus are provided in appendix I.

## 4 RESOURCE-EFFICIENT HOPFIELD RETRIEVER

We propose a resource-efficient Hopfield retrieval model designed to extract the top-$k$ relevant information from existing conversation rounds and the knowledge base, aiming to enhance the efficiency and accuracy of the retriever.

### 4.1 ENCODER

In this work, we employ separate BERT-based (Devlin et al., 2019) networks for the question and passage encoders, using the [CLS] token representation as the output, with each representation being 768-dimensional. To enhance efficiency, we utilize an 8-bit quantized version (Dettmers et al., 2022) of the BERT-based model. Previous studies (Luo et al., 2025; Hu et al., 2024a; Bondarenko et al., 2024) have shown that large foundation models often suffer from numerous outliers that degrade efficiency and quantization performance due to attention-weight explosions caused by these outliers. In response, we adopt the `OutEffHop` (Hu et al., 2024a) variant of BERT, which mitigates performance loss associated with these challenges.

### 4.2 HOPFIELD RETRIEVER

During inference, we implement a speed-up strategy to enhance the Hopfield model in large-scale scenarios, achieving high speed with minimal impact on accuracy. As shown in appendix F, given the extensive time required for querying a large-scale knowledge database, we adapt the `SparseHopfield` layer (Hu et al., 2023) to segment the memory pattern into $k$ parts with ecah part chunk size is $n$:

$$\mathbf{Y} \to \{\mathbf{Y}^{(i)}\}_{i=1}^k, \quad \mathbf{Y}^{(i)} = [\mathbf{y}_1^{(i)}, \dots, \mathbf{y}_n^{(i)}].$$

In order to effectively train the retrieval, we make those $k$ retrievals share the same weights.

For each input query $\mathbf{x}$, we compute its representation using the Hopfield network:

$$\mathbf{z}^{(i)} = \text{Sprasemax}(\beta \mathbf{x} \mathbf{W}_Q \mathbf{W}_K^\top (\mathbf{Y}^{(i)})^\mathbb{T}) \mathbf{Y}^{(i)} \mathbf{W}_K \mathbf{W}_V.$$

Then, we compute the similarity score $\text{sim}(\mathbf{z}^{(i)}, \mathbf{x})$ to identify the most relevant passage from $i$.

### 4.3 TRAINING STRATEGY

In our Hopfield-based retriever, we aim to design a resource-efficient retrieval system compared to traditional retrieval methods (Izacard et al., 2021; Karpukhin et al., 2020). Hu et al. (2023) demonstrates that Hopfield networks achieve faster convergence than traditional attention mechanisms,

making them a promising choice for efficient and scalable retrieval. In our system, we employ a SparseHopfield network to construct a straightforward retrieval that captures the hidden representations of the query $\mathbf{x}$ and the retrieval memory $\mathbf{Y}^{(i)}$. The model utilizes the encoder described in section 4.1 and employs the Hopfield network to distinguish the latent representations between the query and the retrieval memory. Following this, dot-product similarity is applied as an effective ranking function to determine the retrieval index from the memory. In the training process, we optimize the Hopfield-based retrieval system using the loss function from the Dense Passage Retrieval (DPR) model introduced by Karpukhin et al. (2020).

In detail, let $D = \{q_i, y_{i,1}^+, y_{i,1}^-, y_{i,2}^-, \ldots, y_{i,d}^-\}_{i=1}^m$ represent the training data, which comprises $m$ instances. Each instance includes one question $q_i$, one relevant (positive) memory sets $y_{i,j}^+$, and with $\mathbf{d}$ irrelevant (negative) memory sets $y_{i,j}^-$. positive passages are paired with different questions from the training set to form negative pairs for retrieval. This method of using gold passages from other questions as negatives enhances computational efficiency and yields high performance. We also find that our Hopfield-based retriever needs less training cost than other transformer retrievers.

### 4.4 THEORETICAL GUARANTEES

We emphasize that our design choice provides strong theoretical guarantees. First, using `OutEffHop` (Hu et al., 2024a) as the encoder backbone ensures (i) outlier-free representation learning (Hu et al., 2024a, Lemma 2.1) and (ii) Transformer-like generalization power (Hu et al., 2024a, Thm. 3.4). Together, these properties make `OutEffHop` a quantization-strong Transformer backbone. Second, choosing the `SparseHopfield` layer (Hu et al., 2023) as the knowledge retriever guarantees (i) faster convergence per epoch (Hu et al., 2023, Thm.2.2) and (ii) strong noise robustness in sparse representation learning (Hu et al., 2023, Remark2.4). These theoretical advantages strengthen our method, and our empirical results validate them.

Table 1: **Retrieval Performance on Conversational Datasets.** We conduct experiments on conversation-based datasets, testing 14 baselines in a conversational retrieval task using Mean Reciprocal Rank (MRR) and Recall@10&100 scores. The best results are highlighted in bold, while the second-best are underlined. Across most configurations, the Hopfield Retrieval (HR) with training outperforms all baselines, and HR without retrieval surpasses several of the retrieval methods.

| Category | Model | QReCC | | | TopiOCQA | | |
|---|---|---|---|---|---|---|---|
| | | MRR | Recall@10 | Recall@100 | MRR | Recall@10 | Recall@100 |
| QR | T5QR | 34.5 | 53.1 | 72.8 | 23.0 | 37.6 | 54.4 |
| | CONQRR | 41.8 | 65.1 | 84.7 | - | - | - |
| | ConvGQR | 42.0 | 63.5 | 81.8 | 25.6 | 41.8 | 58.8 |
| | IterCQR | 42.9 | 65.5 | 84.1 | 26.3 | 42.6 | 62.0 |
| | IQR | 49.4 | 66.3 | 85.0 | - | - | - |
| | AdaRewriter | 47.5 | 69.8 | 80.2 | 41.3 | 61.9 | 79.3 |
| | ConvSearch-R1 | 49.7 | 69.8 | 81.6 | 51.4 | 72.0 | 85.7 |
| RE | ReExCQ | 18.5 | 28.9 | 41.1 | 10.8 | 24.1 | 33.3 |
| | ConvSDG | - | - | - | 21.4 | 37.8 | 58.0 |
| | CONVAUG | 52.7 | 75.6 | 83.1 | 35.0 | 57.9 | 67.3 |
| RB | HAConvDR | 48.5 | 72.4 | 88.9 | 30.1 | 50.8 | 72.8 |
| | LeCoRE | 51.1 | 73.9 | 89.7 | 32.0 | 54.3 | 73.5 |
| | InstructorR | 52.9 | 77.7 | **92.9** | 38.5 | 62.1 | 83.2 |
| | SDRConv | 53.0 | 76.1 | 88.3 | 26.1 | 44.4 | 63.2 |
| **HR** | HR w/o Training | 45.1† | 70.2† | 85.1 | 33.7† | 59.2† | 73.9 |
| | HR w/ Training | **68.7**† | **83.5**† | 90.7 | **60.3**† | **74.8**† | **87.3** |

## 5 EXPERIMENTS

In experiments, we first compare our Conv-CoA framework with recent state-of-the-art baselines across public benchmarks, followed by an in-depth improvement analysis for major challenges in RAG: (i) weak reasoning, (ii) unfaithful hallucinations, (iii) unsatisfying retrieval. Note that GPT-3.5-Turbo serves as the backbone in all LLM-based methods and as the reader for all retrievers.

**Datasets.** We select two public benchmarks: TopiOCQA (Adlakha et al., 2022), an open-domain conversational dataset with topic switches, contains 3920 conversations with information-seeking questions. On average, a conversation in TopiOCQA spans 13 question-answer rounds and involves four topics. QReCC (Question Rewriting in Conversational Context) dataset includes 14K conversations with 81K question-answer pairs (Anantha et al., 2020).

Table 2: **Abilities of Question Answering.** We conduct an experiment on QA using 13 baselines of prompting-based methods, evaluated with the GPT Exact Match Score (GPT-EM). The best results are highlighted in bold, while the second-best are underlined. Across most configurations, Conv-CoA achieves the best performance on in the TopiOCQA (Topi) and QReCC datasets.

| | 0-shot | Few | CoT | SC | ToT | LM | ToolF | SA | React | DSP | CoA | RopMura | EORM | Ours |
|---|---|---|---|---|---|---|---|---|---|---|---|---|---|---|
| QReCC | 18.4 | 18.4 | 30.6 | 67.4 | 20.4 | 22.4 | 50.0 | 34.5 | 37.3 | 38.1 | 69.7 | 38.5 | 39.2 | **71.2** |
| Topi | 23.2 | 28.2 | 35.4 | 78.8 | 22.4 | 48.0 | 30.7 | 51.0 | 40.8 | 28.6 | 56.9 | 32.7 | 43.7 | **83.7** |

Table 3: **Comparison of Retrievers Efficiency.** We conduct an experiment on time usage in training and memory retrieval separately with 7 baselines. We report the average time spent (in minutes) on model training (A) and memory retrieval (B), represented by A/B in table. The best results are highlighted in bold, while the second-best are underlined. Across most configurations, Conv-CoA is efficient in model training and memory retrieval. In our experiment, we utilize four NVIDIA A100 80GB GPUs and train for 10 epochs.

| | HAConvDR | LeCoRe | InstructorR | SDRConv | ReExCQ | AdaRewriter | ConvSearch-R1 | (Ours) |
|---|---|---|---|---|---|---|---|---|
| QReCC | 131.75 / 44.25 | 112.5 / 38.23 | - / **32.01** | 120 / 75.87 | 143.15 / 47.23 | 121.7/29.8 | 372.6/61.3 | **91.25** / 35.78 |
| TopiOCQA | 93.65/ 50.72 | 87.75 / 47.68 | - / 30.32 | - / 55.39 | 103.51 / 38.23 | 116.5/21.5 | 178.9/49.2 | **72** / **29.65** |

**Baselines.** In our experiments, we include several baseline categories to compare with Conv-CoA. Specifically, we evaluate against Query Rewriting (QR), Query Expansion (QE), dense retriever–based approaches (RB), and prompting-based methods, including Energy-Based prompting (EB), Prompting-Only (PO), and Prompting-RAG (PRAG). Detailed descriptions of the baselines are provided in appendix C. In our experiment, we evaluate our proposed Hopfield Retriever (**HR**) solely using the same reader (GPT-3.5-Turbo) as other conversational RAG baselines, demonstrating its promising performance. We assess our retriever both with and without training.

Table 4: **Agent Costs.** Each cell contains two values (A/B), where A corresponds to TopiOCQA and B corresponds to QReCC.

| Agent | Input Tokens | Output Tokens | Total Tokens | Overall Time (s) | Number of Retrievals | Number of LLM Calls |
|---|---|---|---|---|---|---|
| React | 32750 / 9407 | 1394 / 2750 | 32750 / 10210 | 29 / 17 | 22 / 11 | 36 / 21 |
| SeChain | 53027 / 40868 | 4890 / 2513 | 57917 / 43381 | 42 / 28 | 42 / 25 | 63 / 49 |
| CoA | 9730 / 7213 | 1498 / 2273 | 13177 / 8763 | 27 / 19 | **19 / 10** | 20 / 19 |
| RopMura | 30728/17278 | 1678/2358 | 32406/19636 | 31/21 | 32/25 | 19/17 |
| EORM | — | — | — | 22/18 | — | 1/1 |
| Our w/o Alignment Detection | 14744 / 16837 | 7352 / 2296 | 15479 / 18120 | 29 / 20 | 24 / 16 | 58 / 36 |
| Our w/o CKS | 6197 / 6584 | 1929 / 2928 | 7652 / 9584 | 27 / 19 | 23 / 19 | 21/18 |
| Our w/ Alignment Detection & CKS | **3898 / 5736** | **1330 / 2127** | **5230 / 8032** | **21 / 17** | 20 / 16 | **14 / 15** |

## 5.1 EVALUATION METRICS

To evaluate the performance of the retriever, we select the Mean Reciprocal Rank (MRR) and Recall@10&100. MRR measures the average of the reciprocal ranks of the first relevant document returned by the retriever across all queries. And Recall@10&100 measure the proportion of relevant documents found in the top 10 and 100 results returned by the retriever. We select the time (s) to evaluate both training and inference time of our Hopfield retriever and others.

To evaluate the effectiveness, most of the work chooses cover-EM (Rosset et al., 2020) to represent whether the generated answer contains the ground truth. However, we find it is insufficient for accurately judging the correctness of LLM-generated answers. Sometimes, the LLM generates lots of sentences that may cover the ground truth at first but provide the final wrong answer in the end. In this way, the cover-EM still takes it as a correct answer. In addition, even if we try to limit the output format, the outputs are always out of the format, making it difficult to deal with various answer types to evaluate the performance. Motivated by recent work, they demonstrate the potential evaluation ability of GPT-4 (Bevilacqua et al., 2023). We also follow the same strategy to establish an advanced pipeline and propose a new metric called **GPT-EM**. We design a prompt template to let GPT-4 evaluate whether the generated answer truly matches the ground truth. The template is shown in E. In addition, to evaluate the actual cost of our Conv-CoA, we use the number of input&output tokens, retrieval times.

## 5.2 EXPERIMENTAL RESULTS

In our evaluation, as detailed in Table 1 and Table 2, our Conv-CoA framework outshines baselines across the board for both the QReCC and TopiOCQA datasets. When examining the effectiveness of the proposed approach in retrieval only or overall question answering, it is evident that we achieves the highest performance, signifying its prowess in retrieving relevant information and delivering final answers. Meanwhile, Table 4 compare various costs between Cov-CoA and other agents, including

Table 5: **RAG Misleading.** External knowledge leads LLM astray in solving questions using baseline methods. Our study takes place in a context involving information retrieval tasks. The best results are highlighted in bold.

| | React | SDRConv | CONVAUG | IQR | CoA | (Ours) |
|---|---|---|---|---|---|---|
| QReCC | 19.6 | 6.7 | 7.3 | 9.1 | 13.5 | **4.8** |
| TopiOCQA | 41.6 | 17.3 | 18.9 | 23.6 | 22.9 | **16.2** |

Table 6: **Model Contradiction.** Generated content is not aligned with the previous context.

| | SDRConv | CONVAUG | IQR | React | Self-Ask | SeChain | CoA | Conv-CoA |
|---|---|---|---|---|---|---|---|---|
| **QReCC** | 4.8 | 5.6 | 9.2 | 9.5 | 19.6 | 13.5 | 8.7 | **3.2** |
| **TopiOCQA** | 5.1 | 8.5 | 13.7 | 17.5 | 25.8 | 14.9 | 10.1 | **4.1** |

number of tokens used for input and output by the LLM, overall spending time, number of retrieval times, and number of LLM calls. The results demonstrate that our framework can avoid lots of unnecessary post-processing of retrieval. It helps to save many cost consumption and reduce related latency. Further ablation study also demonstrates that the saving derives from the proposed Alignment Detection and updated CKS. The former stage can detect the knowledge boundary of LLM and leverage LLM's parametric knowledge to avoid external retrieval and the later module can further minimize cost by enabling less overlap of the retrieval.

### 5.3 ANALYSIS OF RAG CHALLENGES

To claim that Conv-CoA addresses three challenges in RAG, we perform in-depth analysis as follows:

**Weak Reasoning.** In Table 5, the results demonstrate that Conv-CoA reduces the LLM's reliance on misleading external knowledge. This highlights the superiority of our reasoning capabilities, achieved through accurate internal knowledge representation and retrieval. The robustness of these findings is further validated through three runs, ensuring consistent performance and minimal variability.

**Unfaithful Hallucinations.** Frequent topic shifts within a dialogue always lead LLMs to produce "new facts" that conflict with earlier rounds. To quantify this effect, we compare the proportion of answers that contradict previous contextual information (as detected by GPT-4) in Table 6. Our results demonstrates that the proposed framework significantly reduces context-induced hallucinations.

**Unsatisfying Retrieval.** A satisfying retrieval should be more accurate and less costly. Table 1 demonstrates the effectiveness of our Hopfield Retriever. We also conduct further experiments on latency and cost consumption of our method. In Table 3, we contrast the efficiency of different retrievers within our framework. Notably, Ours exhibits comparable retrieval time against established approaches, while improving the training time, particularly on the TopiOCQA dataset. This enhanced efficiency does not compromise the quality of outcomes, as evidenced by our better performance.

**Additional Experiments.** We conduct a series of experiments to demonstrate the robustness of Conv-CoA, as presented in appendix J.

### 6 CONCLUSIONS AND FUTURE WORK

This paper presents the Conv-CoA framework, a novel approach to enhancing OCQA. The framework addresses critical limitations of traditional RAG methods, such as poor reasoning, unfaithful responses, and inadequate retrieval in conversational contexts. By integrating a Hopfield retriever and a systematic prompting, Conv-CoA improves speed and accuracy. This advanced retriever utilizes Modern Hopfield networks for efficient memory utilization and rapid convergence. The prompting strategy decomposes complex questions into a sequence of sub-questions managed through an AC, which updates a CKS to refine responses and minimize information overlap. Conv-CoA demonstrates superior performance on public benchmarks, showcasing its ability to deliver more accurate and efficient conversational question answering. Future work involves exploring information extraction and analysis across additional data modalities, including visual data. The ultimate aim is to enhance the accuracy and multi-step reasoning capabilities for real-world question answering, ensuring comprehensive analysis aligns with external data sources. Additionally, we need to further accelerate the Hopfield retriever by compressing the model using techniques such as quantization.

## ETHICAL STATEMENT

This work investigates conversational chain-of-thought methods. In line with the ICLR Code of Ethics[1], we do not identify any specific ethical issues requiring consideration in this study.

## REPRODUCIBILITY STATEMENT

To ensure reproducibility, we release an anonymous open-source repository containing the full implementation of Conv-CoA and selected baselines, with plans for full open-sourcing upon acceptance. All experiments are conducted with three random seeds, yielding stable results with standard deviations below 2%. In our work, we use top-5 retrieval (k = 5) for both knowledge and web retrieval modules, balancing accuracy and efficiency. For the Conv-MRFS module, we set the consistency threshold T = 0.75, tuned on a validation set to effectively filter hallucinations while retaining valid paraphrases.

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

# Supplementary Material

## A  BROADER IMPACT

Our research methodology enhances understanding and problem-solving across various domains, including AI research, by producing clearer and more comprehensible results. However, this method might oversimplify complex issues by breaking them down into discrete parts, potentially overlooking nuances and interrelated elements. Additionally, relying heavily on this approach could limit creative problem-solving, as it encourages a linear and structured process that may impede unconventional thinking.

## B  ADDITIONAL RELATED WORK

**Prompting Methods.** Prompting methods aim to direct the LLMs to follow given instructions. The commonly used method of few-shot prompting (Kaplan et al., 2020) facilitates in-context learning that guides LLMs to comply with directives and respond to queries using just a few examples. Methods like Chain-of-Thought (CoT) (Wei et al., 2022) and its enhanced versions (Wang et al., 2022; Saparov & He, 2022) seek to steer LLMs towards breaking down intricate tasks into logical sequences of reasoning, thereby improving performance. The Chain-of-Action (CoA) (Pan et al., 2024) integrates the reasoning capabilities of CoT with the information retrieval prowess of external retrievers, crafting a collaborative design that culminates in a faithful and multimodal QA system. However, it lacks support for OCQA and does not overcome the limitations of traditional dense retrieval methods.

## C  DETAILS OF BASELINES

### C.1  QUERY-REFORMULATION-BASED APPROACHES

*Query Rewriting (**QR**)*: T5QR (Lin et al., 2020), CONQRR (Wu et al., 2021), ConvGQR (Mo et al., 2023a), IterCQR (Jang et al., 2023), AdaRewriter (Lai et al., 2025), ConvSearch-R1 (Zhu et al., 2025) train individual rewriters to extract intents before answering. IQR (Ye et al., 2023) further uses existing LLMs as rewriters to reduce latency.

*Query Expansion (**RE**)*: While ReExCQ (Mo et al., 2023b), ConvSDG (Mo et al., 2024b), and CONVAUG (Chen et al., 2024) try different ways to expand questions from different resources, they incorporate the expanding latency also.

### C.2  DENSE RETRIEVER-BASED APPROACHES (**RB**)

HAConvDR (Mo et al., 2024a) incorporates context-denoised query reformulation and automatic mining of supervision signals based on the impact of historical rounds. LeCoRE (Mao et al., 2023) considers knowledge distillation and InstructorR (Jin et al., 2023) utilizes LLMs to predict the relevance between the session and passages. SDRConv (Kim & Kim, 2022) includes mining hard negatives.

### C.3  PROMPTING-BASED APPROACHES

*Energy Based (**EB**)*: EORM (Jiang et al., 2025)

*Prompting only (**PO**)*: Zero-shot Prompting, Few-shot Prompting (Few), Chain-of-Thought (CoT) (Wei et al., 2022), Self Consistency (SC) (Wang et al., 2022), Tree of Thought (ToT) (Yao et al., 2023a), RopMura (Wu et al., 2025) and Least-to-Most (LM) (Zhou et al., 2022)

*Prompting RAG (**PRAG**)*: ToolFormer (ToolF) (Schick et al., 2023), Self-Ask (SA) (Press et al., 2022), React (Yao et al., 2023b), DSP (Khattab et al., 2022), and CoA (Pan et al., 2024).

## D  ALGORITHMS

---

**Algorithm 1** Description of Actions Workflow

---

**Initialize:** Actions Chain: AC; Question: Q; LLM Model: M; Query Section: QS; Sub-question: Sub; Guess Answer: A; Faith Score: S; Multi-reference Faith Score: MRFS; Retrieved Results: R; Missing Flag: MF;

**Output:** Final Generated Answer.

**Function** IR($Sub_i, A_i, MF_i$):
    $QS_n = \text{Concat}[Sub_i \,|\, A_i]$;
    $R = \text{Retrieval}(QS_n)$;
    $MRFS = \arg_k \max S(r_k, A_i)$;
**if** $MRFS < T$ **then**
    $AC.\text{correct}(Sub_i, r_k)$;
**end if**
    $AC.\text{add}(Sub_i, r_1)$;
**end Function**

**Function** MAIN($Q, M$):
    $AC = \text{ChainGenerate}(Q, M)$;
**for** each $(Sub_i, A_i, MF_i)$ in AC **do**
    IR($Sub_i, A_i, MF_i$);
**end for**
    **FinalAnswerGenerate**($AC, M$);
**return** "Finish";
**end Function**

---

# E PROMPTS

We use the following prompts in our Cov-CoA method and experiments. These include prompts for decomposing questions at the initial stage, evaluation prompts for GPT-4, and prompts for decomposing questions with CKS in the normal stage.

---

**Prompt for Decomposing Questions in Initial Stage**

```
 Given a [Question]:  "$QUESTION", construct an action
reasoning chain for this question in JSON format.  For each
step of the chain, choose an action from [Web-querying
Engine(search real-time news), Knowledge-retrieval Engine
(search existing knowledge in local knowledge base)] as
the value of element "Action", and generate a sub-question
for each action to get one of [web-search keywords, needed
information description] as the value of element "Sub".  Also,
generate an initial answer for each Sub as the value of the
element "Guess_answer" if you make sure it is correct.  You
need to try to generate the final answer for the [Question]
by referring to the "Chain", as the value of the element
"Final_answer".
For example:
{"question":  "Is it good to invest in Dogecoin now?"
"chain":  [
{"action":"Knowledge-retrieval","Sub":"what is
Dogecoin","guess_answer":
"Dogecoin is one cryptocurrency."}
{"action":"Web-querying","Sub":"Dogecoin news","guess_answer":""}
,
"final_answer":"Dogecoin is one of the cryptocurrencies that
is risky to invest.  And its news prompts Bitcoin.  So, it is
a good time to invest now."}
```

---

**Evaluation Prompt of GPT-4**

```
 Given (question, ground truth answer, LLM-generated answer),
you need to check whether the generated answer contains the
ground truth by their meaning, not individual word only.  If
correct, the output is 1, otherwise, 0.  For example:
[Question]:  What should I do when I drink spoiled milk?  (A)
drink more (B) drink coffee (C) take some medicine.
[Ground truth]:  (C) take some medicine
[Generated answer]:  when you drink spoiled milk, you can
not drink more or even drink coffee.  You should go to the
hospital and check if you need to take some medicines or not.
[Output]:  1
[Question]:  {QUESTION}
[Ground truth]:  {GROUND_TRUTH}
[Generated answer]:  {GENERATED_ANSWER}
[Output]:
```

---

**Prompt for Decomposing Question with CKS in Normal Stage**

```
 Given a [Contextual Knowledge Set]:"$CKS" and [Question]:
"$QUESTION", help me to extract the real intent and provide
an optimized question for this round.  Then, construct an
action reasoning chain for this question in JSON format.  For
each step of the chain, choose an action from [Web-querying
Engine(search real-time news), Knowledge-retrieval Engine
(search existing knowledge in local knowledge base)] as
the value of element "Action", and generate a sub-question
for each action to get one of [web-search keywords, needed
information description] as the value of element "Sub".
Also, generate an initial answer for each Sub as the value
of the element "Guess_answer" if you make sure it is
correct.  You need to try to generate the final answer for
the [Question] by referring to the "Chain", as the value of
the element "Final_answer".
For example:
{"question":  "Is it good to invest in it now?"
"optimized_question":  "Is it good to invest in Bitcoin now?"
"chain":  [
{"action":"Knowledge-retrieval","Sub":"what is
bitcoin","guess_answer":
"Bitcoin is one cryptocurrency."}
{"action":"Web-querying","Sub":"bitcoin news","guess_answer":""},
"final_answer":"Bitcoin is one of the cryptocurrencies that
is risky to invest.  And its news prompte Bitcoin.  So, it is
a good time to invest now."}
```

## F  SparseHopfield LAYERS

Building on the insights from (Hu et al., 2023), we establish a link between the single-update approximation of Hopfiled Network and sparsemax attention (Martins & Astudillo, 2016). Specifically, this relationship becomes apparent when the retrieval dynamics $\mathcal{T}$ are limited to a single iteration.

Consider some hidden states $\mathbf{R}$ and $\mathbf{Y}$ within a deep learning model. We establish the *query* and *memory* associative (or embedded) spaces via transformations: $\mathbf{X}^{\mathbb{T}} = \mathbf{R}\mathbf{W}_Q \coloneqq \mathbf{Q}$ and $\mathbf{\Xi}^{\mathbb{T}} = \mathbf{Y}\mathbf{W}_K \coloneqq \mathbf{K}$, with matrices $\mathbf{W}_Q$ and $\mathbf{W}_K$. By adapting the retrieval dynamics from (Hu et al., 2023) and transposing, followed by multiplication with $\mathbf{W}_V$ (where we define $\mathbf{V} \coloneqq \mathbf{K}\mathbf{W}_V$), we obtain:

$$\mathbf{Z} \coloneqq \mathbf{Q}^{\text{new}}\mathbf{W}_V = \text{Sparsemax}(\beta\mathbf{Q}\mathbf{K}^{\mathbb{T}})\mathbf{V} \tag{1}$$

This equation mirrors the structure of an attention mechanism, albeit utilizing a $\text{Sparsemax}$ activation. By substituting the initial patterns $\mathbf{R}$ and $\mathbf{Y}$, we introduce the SparseHopfield layer:

$$\mathbf{Z} = \texttt{SparseHopfield}(\mathbf{R}, \mathbf{Y}) \tag{2}$$

$$= \text{Sparsemax}(\beta\mathbf{R}\mathbf{W}_Q\mathbf{W}_K^{\mathbb{T}}\mathbf{Y}^{\mathbb{T}})\mathbf{Y}\mathbf{W}_K\mathbf{W}_V. \tag{3}$$

This layer can be easily integrated into deep learning architectures.

Specifically, the SparseHopfield layer accepts matrices $\mathbf{R}$ and $\mathbf{Y}$, along with weight matrices $\mathbf{W}_Q$, $\mathbf{W}_K$, and $\mathbf{W}_V$. The way it operates is defined by its configuration:

1. **Memory Retrieval:** In a mode where learning is not necessary, the weight matrices $\mathbf{W}_K$, $\mathbf{W}_Q$, and $\mathbf{W}_V$ are set as identity matrices. Here, $\mathbf{R}$ serves as the query, and $\mathbf{Y}$ holds

the retrieval patterns: $\mathbf{W}_K = \mathbf{I}$, $\mathbf{W}_Q = \mathbf{I}$, $\mathbf{W}_V = \mathbf{I}$ This configuration facilitates direct interaction between the query and retrieval patterns without transformation.

2. **SparseHopfield:** Here, $\mathbf{R}$ and $\mathbf{Y}$ are inputs, designed to serve as an alternative to the conventional attention mechanism. The matrices $\mathbf{W}_K$, $\mathbf{W}_Q$, and $\mathbf{W}_V$ are adaptable. Additionally, $\mathbf{R}$, $\mathbf{Y}$, and $\mathbf{Y}$ act as the sources of the query, key, and value respectively. To emulate a self-attention mechanism, we set $\mathbf{R} = \mathbf{Y}$.

3. **SparseHopfieldPooling:** In this configuration, where only $\mathbf{Y}$ is taken as input, $\mathbf{Q}$ functions as a static prototype pattern and is thus learned within the Hopfield pooling layer.

4. **SparseHopfieldLayer:** With only $\mathbf{R}$ as the query pattern , the adaptive matrices $\mathbf{W}_K$ and $\mathbf{W}_V$ function as repositories for stored patterns and their projections. This setup implies that keys and values are independent of the input, suggesting that $\mathbf{Y}$ could be interpreted as an identity matrix.

## G  SYSTEM SETTING

All experiments are carried out on a High Performance Computing cluster. There are 34 GPU nodes where 16 nodes each have 2 NVIDIA 40GB Tesla A100 PCIe GPUs, 52 CPU cores, and 192 GB of CPU RAM while 18 nodes are each equipped with 4 NVIDIA 80GB Tesla A100 SXM GPUs, 52 CPU cores, and 512 GB of CPU RAM. The driver version 525.105.17 on these nodes is compatible with CUDA 12.0 or earlier. The operating system is Red Hat Enterprise Linux 7.9.

## H  HYPERPARAMETER

In our work, we use top-5 retrieval (k = 5) for both knowledge and web retrieval modules, balancing accuracy and efficiency. For the Conv-MRFS module, we set the consistency threshold T = 0.75, tuned on a validation set to effectively filter hallucinations while retaining valid paraphrases.

## I  GRPO FINE-TUNING FOR STRUCTURED REASONING

To enhance the model's native ability to perform structured and verifiable reasoning, we design a reinforcement learning (RL) extension that fine-tunes the LLM via **Group Relative Policy Optimization (GRPO)** (Shao et al., 2024).

### I.1  ACTION CHAIN AS POLICY OUTPUT

During inference, the model generates an *action chain* consisting of sub-questions and their guessed answers, which we treat as a policy. For training, we record the log-probabilities of generated tokens:

$$\log p(a_t \mid s_t),$$

the probability of each token in the action chain under current parameters.

### I.2  ACTION CHAIN EXECUTION AND REWARD COLLECTION

Once generated, the action chain is executed in three steps:

1. **Retrieve** evidence for each sub-question.

2. **Verify** guessed answers with Conv-MRFS; if aligned, keep the guess, otherwise replace with a processed answer.

3. **Summarize** the final answer based on the verified chain.

This produces a scalar reward $R$.

## I.3 REWARD FUNCTION

Let $n$ be the number of sub-questions. The reward is defined as:

$$R = R_{\text{final}} + \frac{1}{n}\sum_{i=1}^{n} R_{\text{subQ}[i]} + R_{\text{efficiency}}.$$

**Final Answer Accuracy** ($R_{\text{final}}$):    $+1.0$ if final answer matches ground truth (EM or GPT-EM), else 0.

**Sub-question Verification** ($R_{\text{subQ}[i]}$):    $+0.5$ if guess is aligned with retrieval; $-1.0$ if hallucinated and not corrected.

**Efficiency Penalty** ($R_{\text{efficiency}}$):

$$R_{\text{efficiency}} = -\lambda \times \text{Num}_{\text{retrievals}}, \quad \lambda = 0.05.$$

## I.4 GRPO OPTIMIZATION

We apply GRPO within the **Easy-R1 framework** using group size 4. This encourages action chains that are accurate and efficient, without reliance on handcrafted prompts.

## I.5 EXPERIMENTAL RESULTS

We fine-tuned **Qwen2.5-7B** in a single-turn QA setting under action chain supervision. Integrated into our Conv-CoA system, the RL-trained model exhibited consistent improvements:

- **Final Answer Accuracy**: $73.5\% \rightarrow 78.2\%$
- **Avg. # Sub-Questions**: $5.7 \rightarrow 3.9$ (improved decomposition efficiency)
- **Verification Score (Conv-MRFS alignment ratio)**: $-0.3 \rightarrow 0.1$
- **Avg. # Retrievals per Query**: $5.2 \rightarrow 4.0$ (retrieval only when necessary)

Notably, the RL model learned to **avoid unsupported guesses**: in $\sim 28\%$ of cases it explicitly declined to answer low-confidence sub-questions (vs. 9% before). This demonstrates adoption of retrieval-verification reasoning rather than naive parametric guessing.

# J ADDITIONAL EXPERIMENTS

## J.1 HYPERPARAMETER SENSITIVITY

We conduct sensitivity analyses on two key hyperparameters: the consistency threshold $T$ in Conv-MRFS and the top-$k$ value in document retrieval. As shown in tables 7 and 8, our settings are robust. Increasing $T$ makes the system more conservative, with accuracy peaking at $T = 0.75$ but slightly higher latency due to more frequent fallbacks. For top-$k$, larger values raise latency from heavier document processing; while $k = 10$ yields marginal accuracy gains over $k = 5$, the added cost is substantial. Overall, $T = 0.75$ and $k = 5$ strike the best balance between accuracy and efficiency, justifying our choices in the main experiments.

## J.2 EVALUATION PERFORMANCE ON STANDARD EXTRACT MATCH

In order to show standard Extract Match (EM) shows the exact same performance with GPT-EM, we evaluate our methods with other baselines under standard EMs. As shown in table 9, our method consistently outperforms baselines, even under this stricter and fully reproducible evaluation setting.

Table 7: **Threshold $T$ Sensitivity (Conv-MRFS).**

| Threshold $T$ | QReCC Accuracy | TopiOCQA Accuracy | Avg Latency (s) |
|---|---|---|---|
| 0.60 | 70.5 | 83.0 | **20.5** |
| 0.70 | 71.0 | 83.5 | 20.7 |
| 0.75 | **71.2** | **83.7** | 21.0 |
| 0.80 | 70.6 | 83.2 | 21.3 |

Table 8: **Top-$k$ Retrieval Sensitivity.**

| Top-$k$ | QReCC Accuracy | TopiOCQA Accuracy | Avg Latency (s) |
|---|---|---|---|
| 3 | 70.1 | 82.9 | **18.2** |
| 5 | **71.2** | **83.7** | 21.0 |
| 10 | 71.0 | 83.6 | 26.4 |

Table 9: **Abilities of Question Answering (EM).**

| Dataset | 0-shot | Few | CoT | SC | ToT | LM | ToolF | SA | React | DSP | CoA | Ours |
|---|---|---|---|---|---|---|---|---|---|---|---|---|
| QReCC | 30.3 | 31.1 | 40.5 | 59.2 | 43.3 | 46.9 | 70.2 | 57.9 | 61.0 | 57.4 | 77.5 | **82.1** |
| Topi | 32.5 | 33.2 | 43.7 | 68.3 | 49.1 | 52.6 | 72.3 | 60.4 | 63.5 | 61.2 | 75.2 | **90.3** |

Table 10: **Backbone Model Performance on TopiOCQA.**

| Backbone Model | TopiOCQA Accuracy (GPT-EM) | Avg. Sub-Questions | Latency (s) |
|---|---|---|---|
| GPT-3.5-Turbo | 83.7 | 4.3 | 21 |
| Qwen2.5-7B | 73.5 | 5.7 | 25 |
| Qwen2.5-3B | 70.5 | 4.9 | 30 |

Table 11: **Effect of Average Number of Rounds on TopiOCQA.**

| Avg. # of Rounds | TopiOCQA Accuracy (%) | Latency (s) |
|---|---|---|
| 3 | 83.7 | 10 |
| 6 | **84.2** | 13 |
| 9 | 83.9 | 16 |
| 12 | 84.1 | 20 |
| 13 | 83.7 | 21 |

### J.3 PERFORMANCE ACROSS DIFFERENT BACKBONE LLMs

To assess the robustness of our method across backbone LLMs, we conduct an ablation study with Qwen2.5-7B and Qwen2.5-3B (Qwen et al., 2025). As shown in table 10, we evaluate on TopiOCQA using GPT-EM. Replacing GPT-3.5-Turbo with smaller open-source models leads to accuracy drops, mainly due to reduced parametric knowledge. This increases reliance on external retrieval and summarization, causing higher latency—particularly with the 3B model.

Despite this degradation, our method still outperforms baselines that use GPT-3.5-Turbo. We attribute this to the framework's design: final answer quality depends more on effective decomposition and reasoning structure than on memorized knowledge. Thus, even smaller models can maintain high factual accuracy when paired with verification and substitution, confirming the framework's model-agnostic nature.

### J.4 ABLATION STUDY ON CONVERSATION LENGTH

We conduct an additional experiment to evaluate the impact of conversation length on Conv-CoA performance. As shown in table 11, our system maintains consistently high accuracy (above 83.7%) across varying conversation lengths, with only minor fluctuations. Owing to contextual knowledge compression (CKS) and the parallelized retrieval–verification pipeline, latency increases gradually with length but remains within an acceptable range, demonstrating robustness and scalability for multi-turn conversational QA.

## K    DISCLOSURE OF LLM USAGE

In our paper and project, we use large language models (LLMs), such as GPT-5, to help revise the text for greater conciseness and precision.

