# OpenReview forum: "Conv-CoA: Open-domain Question Answering via Conversational Chain-of-Action with Hopfield Retriever"
_ICLR.cc/2026/Conference — ICLR 2026 Conference Withdrawn Submission_

### Official Review · Reviewer_LcLE · 2025-10-30

**Soundness:** 2
**Presentation:** 2
**Contribution:** 2
**Rating:** 2
**Confidence:** 4

**Summary:**

This paper presents a new system for multi-turn conversational question answering. At every step of the conversation, the system first generates a "chain of action" (CoA) which consists of a sequence of retrieval actions and question answering actions. The retrieval actions are used to fetch relevant documents from a Hopfield-based retriever, while the QA actions generate intermediate answers or the final answer. The CoA is generated using a large language model (LLM) prompted with the conversation history and previous CoA steps. The authors utilize Hopfield networks to improve training and inference time efficiency of the retriever.

Experiments on two conversational QA datasets (QReCC, TopicOCQA) show that Conv-CoA outperforms competitive methods, and is more efficient in terms of retrieval time and LLM tokens.

**Strengths:**

1. The paper works on an important and timely problem: multi-turn conversational question answering. These days LLMs are being used extensively in multiturn educational setups, customer support etc., which makes conversational QA an important research area.

2. The paper has an interesting idea to use Hopfield networks for efficient retrieval.

**Weaknesses:**

1. **Baselines and experimental setup is quite outdated**: The paper currently uses GPT3.5 as a backbone, which is a model from 2022. LLMs have gotten a lot better since then, with significant improvements in zero-shot factuality, long-context, and reasoning with thinking tokens. Given this, I'm not very sure the loss patterns being tackled in the paper are valid anymore with SoTA models, and I expect they will do a lot better. Similarly, the retriever uses BERT encoders as backbones, which is a model from 2018.

2. **Paper would be stronger with evaluation on more recent / popular benchmarks**: The two datasets used in the paper (QReCC, TopicOCQA) are both quite old and not as popular as QuAC / CoQA in conversational QA. The paper would be much stronger with evaluation on recent multiturn LLM benchmarks like Scale's MultiChallenge (https://arxiv.org/abs/2501.17399) which is more reflective of current LLM users. Similarly, the hopfield retriever should be evaluated on MTEB (https://github.com/embeddings-benchmark/mteb).

3. **The method seems overly complicated, and I'm a bit confused by some design decisions**: There are several moving parts in the proposed algorithm, and many of these may not be needed with a stronger LLM backbone like GPT5. I was also not able to understand Section 3.3/3.4.2 and how it fits into Fig 1. Why is alignment detection needed between guessed answer and context? Can't all the results be directly fed into long context of the model to avoid a guesswork stage? The only reason I can see is L261: "avoid unnecessary post-processing of retrieval and reduce the hallucinations.", but I don't see modern LLMs which use thinking tokens making such errors. Similarly why is summarization of the snippeted needed in Section 3.4.1.

4. **The paper is lacking in qualitative examples and analysis**: Currently, it's a bit hard to understand the exact types of problems the proposed methodology is tackling, and type of prompts the model gets better at using this system. The paper would be a lot stronger with a qualitative analysis of wins and losses between Conv-CoA and vanilla RAG baselines.

5. Minor: I think the Hopfield network needs more justification. How does Hopfield compare to simpler methods like FAISS on dense embeddings which also save inference time? Why is retriever training time a bottleneck when it's <2 hrs?

**Questions:**

See weaknesses ^

---

### Official Review · Reviewer_y1tH · 2025-10-31

**Soundness:** 1
**Presentation:** 1
**Contribution:** 1
**Rating:** 2
**Confidence:** 4

**Summary:**

The paper introduces Conv-CoA, a framework for open-domain conversational question answering (OCQA). It proposes decomposing complex questions into sub-questions through a “chain-of-action” structure, combining two predefined actions (web-querying and knowledge-retrieval) and a Hopfield-based retriever. The framework maintains a Contextual Knowledge Set (CKS) to track conversation state and introduces a Conversational Multi-Reference Faith Score (Conv-MRFS) to detect answer–retrieval inconsistencies. Experiments on QReCC and TopiOCQA claim higher accuracy and efficiency compared to previous retrieval-augmented generation (RAG) and prompting methods.

**Strengths:**

- Addresses real issues in conversational QA, including hallucination, retrieval latency, and reasoning over dialogue history.
- The idea of decomposing complex questions into sub-questions is appealing and intuitively aligns with current research on reasoning chains and agentic workflows.

**Weaknesses:**

- Lack of awareness of recent work. The paper cites 2021–2023 systems (e.g., CONQRR, ReExCQ) as “recent,” overlooking 2024–2025 advances in agentic RAG, retrieval-augmented LLMs, and tool-use frameworks.

- Writing and structure. The exposition is difficult to follow: core tasks (reasoning, retrieval, alignment) are not clearly defined, terminology like “unprepossessing” is unclear, and the narrative mixes motivation, method, and implementation details without flow.

- Questionable novelty. The “two predefined actions (web-querying and info-searching) are standard in multi-tool QA systems.

- Lack of details. The use of a Hopfield retriever is novel in name but under-explained; the link between Hopfield physics and retrieval efficiency is asserted rather than demonstrated.

- Methodology unclear. Many details are missing, e.g. how Conv-MRFS is computed and how the “alignment” signal feeds back into reasoning are hard to parse. The “plus” RL version is only briefly mentioned in the appendix without integration into the main text. Also, what do you mean by “unprepossessing”?

- Experimental setup. Results are based on small, outdated QA datasets (TopiOCQA, QReCC) and pre-LLM baselines. Modern LLMs could process entire contexts directly, reducing the relevance of these comparisons. Tables are dense and barely readable, and it not clear what metrics is used in tables 5 and 6; metric definitions (e.g., GPT-EM) are vague (what is the difference with llm-as-judge?).

- Overall presentation. The paper should be substantially rewritten to clarify objectives, provide concrete algorithmic details, and explain the retrieval mechanism and decision logic (e.g., when to invoke web vs. knowledge retrieval).

**Questions:**

- What is the actual intuition behind the Hopfield retriever? How does it address issues of existing methds?
- How is Conv-MRFS trained or tuned? What data supports its precision/recall components?
- Why were QReCC and TopiOCQA chosen given newer conversational QA benchmarks and stronger LLM baselines?
- How is the paper contributing to current state of the art in CONVQA?

---

### Official Review · Reviewer_6Tcd · 2025-11-01

**Soundness:** 3
**Presentation:** 3
**Contribution:** 2
**Rating:** 2
**Confidence:** 4

**Summary:**

This paper introduces Conv-CoA, a framework for Open-domain Conversational Question Answering. It addresses challenges like unfaithful hallucination, weak reasoning, and unsatisfying retrieval by using a dynamic reasoning-retrieval mechanism with a novel Hopfield-based retriever and a faith score to verify answers. Conv-CoA outperforms other methods in accuracy and efficiency on public benchmarks.

**Strengths:**

The paper proposes an efficient and effective method for incorporating RAG with open domain QA.  The strengths are:
1. The proposed method is simple but effective
2. The paper's writing is neat
3. sufficient experiments

**Weaknesses:**

The technical contribution is the major concern. The work has three main new parts: 1. split the overall question into sub-questions and form the CKS; 2. adopt the hopfiled reteirver to have more efficient retrieval. 3. The faith score;  However, I think the novelty is marginal.

Also, there are parts that are unclear:
1. RAG + open QA is not new, had a quick search with keywords RAG for open QA, there are a lot of papers listed. Could consider adding some description about this group of works.
2.  It is better to add more description (rather than architecture) on why the hopfiled reteirver could help on the efficiency of retrieval ( we also find that Table 3 shows it is not the most efficient method). It is suggested to add more description of memory management and the size.
3. 4.4 does not look like a theoretical guarantee.  It is just an explanation.
4. Although cover-EM has some issues, it would be better to also add its results or case study, so that it would be clearer of its benefit is compared to GPT-EM
5.  It is better to have more descriptions of the experiments in 5.3.

**Questions:**

The questions are releted to the weakness

1. could you add more RAG + open QA works?
2. why the hopfiled reteirver could help on the efficiency of retrieval? why it is not the most efficient one in the table?
3. how to manage the memory?
4.  could you add results using cover-EM?
5. how 5.3 experiments are setup? espeically the first two?

---

### Official Review · Reviewer_aVTK · 2025-11-04

**Soundness:** 2
**Presentation:** 3
**Contribution:** 2
**Rating:** 4
**Confidence:** 3

**Summary:**

The paper introduces Conv-CoA (Conversational Chain-of-Action), a framework for open-domain conversational question answering (OCQA). It aims to overcome three persistent challenges in retrieval-augmented generation (RAG) systems: 1/ Unfaithful hallucinations, with responses inconsistent with domain or real-time facts, 2/ weak reasoning, large language models (LLMs) fails to extract or combine relevant information, 3/ unsatisfying retrieval, for example traditional dense retrievers not capturing conversational intent. The proposed approach decomposes complex conversational questions into a chain of sub-questions, each verified through retrieved data and faithfulness scoring. Experiments on QReCC and TopiOCQA show that Conv-CoA outperforms SOTA baselines while reducing hallucinations and retrieval cost.

**Strengths:**

The paper presents novel ideas like the integration of Hopfield-based retrieval with a chain-of-action reasoning framework, which contribute to bridge reasoning and retrieval. The performances achieved support this intuition and the empirical evaluation is comprehensive and exhausting.

**Weaknesses:**

The paper tends to lack in the level of details provided. For example, the explanation of each component is shallow and presented with limited theoretical or schematic detail. There are parts that are complex and it is unclear why such complexity is required. For example, the pipeline is quite complex and design choices are not well document (for instance, why using different prompt templates to generate sub-questions?). Finally, from the appendix results on backbone LLMs, it appears that performances drop when using smaller model. I would encourage the authors to dive deeper into this, and provide intuitions on why the success of this approach is not entirely due to the large size of the LLM used as a backbone.
Minor comment: multimodal applicability is mentioned but not explored or demonstrated.

**Questions:**

see above

---

### Note · Authors · 2025-11-12

**Comment:**

Dear AC, SAC, PC, and Reviewers,

Thank you all for taking the time to carefully review our manuscript. We have thoroughly read your valuable comments and feedback. After careful consideration, we have decided to withdraw this submission to allow sufficient time for substantial revisions. We sincerely appreciate your insightful suggestions, which will greatly help us improve the work. We look forward to resubmitting a significantly improved version in the future.

Thank you again for your time and support.

Best regards,
Authors

**Withdrawal Confirmation:**

I have read and agree with the venue's withdrawal policy on behalf of myself and my co-authors.